# Facilitating Factors and Barriers in the Return to Work of Working Women Survivors of Breast Cancer: A Qualitative Study

**DOI:** 10.3390/cancers15030874

**Published:** 2023-01-31

**Authors:** Rebeca Marinas-Sanz, Isabel Iguacel, Jerónimo Maqueda, Laura Mínguez, Paula Alquézar, Raquel Andrés, Esther Pérez, Ramón Sousa, Elena Moreno-Atahonero, Dolors Solé, Antonio Güemes, Begoña Martínez-Jarreta

**Affiliations:** 1Scientific Research Group GIIS-063 of Aragon Institute of Health, Department of Occupational Medicine, University of Zaragoza, 50009 Zaragoza, Spain; 2Faculty of Health Sciences, University of Zaragoza, 50009 Zaragoza, Spain; 3Department of Health Promotion and Occupational Epidemiology, National Institute of Occupational Safety and Health at Work (INSST), 28027 Madrid, Spain; 4Breast Cancer Unit, “Lozano Blesa” Clinical University Hospital, 50009 Zaragoza, Spain

**Keywords:** breast cancer, qualitative study, focus groups, sick leave, work, return to work

## Abstract

**Simple Summary:**

Working breast cancer survivors face several challenges when returning to work (RTW). Some qualitative studies have been carried out taking into account just the vision of these women in relation to the facilitators and barriers encountered when RTW. However, this study aims to complement the patient’s vision with two very active agents in this process, such as health professionals and company managers. Although the experience of working women with breast cancer, health professionals and company managers did not always coincide, these focus groups provided many barriers and facilitators in the RTW process of female workers with breast cancer. Many of the problems seen by these agents were related to the lack of understanding and adaptation to the situation (e.g., the company did not provide an explanation to the colleagues of the affected worker). New protocols and multidisciplinary work involving different agents (including physical and psychological rehabilitation) seem to be necessary.

**Abstract:**

Several studies have identified the main barriers and facilitators that breast cancer survivors experience in the return to work (RTW). The authors conducted a qualitative study using focus group discussions with a group of female non-metastatic breast cancer survivors (n = 6), a group of health professionals from different medical specialties (n = 8), and a third group of company managers mainly composed of human resources managers (n = 7). The study was carried out between March and December 2021 in Zaragoza (Spain). Transcripts were analyzed using inductive content analysis to identify work-related barriers and facilitators and coded by the research team. Barriers identified included physical and cognitive symptoms, psychosocial problems, lack of knowledge and coordination (health professional, patients, and managers), legal vacuum, physical change, time constraints, work characteristics (lower skilled jobs), unsupportive supervisors and coworkers, family problems and self-demand. Facilitators included family and work support, physical activity and rehabilitation, personalized attention, interdisciplinary collaboration, legal advice for workers, knowledge about breast cancer in companies, positive aspects of work, elaboration of protocols for RTW in women with breast cancer. RTW in working women with breast cancer requires a personalized and holistic view that includes the perspectives of patients, healthcare professionals and company managers.

## 1. Introduction

Breast cancer is the most diagnosed cancer type worldwide. In 2020, approximately there were 2.3 million new cases of breast cancer globally and about 685,000 deaths from this disease [1]. In Spain 34,088 new cases of breast cancer were diagnosed in 2020 according to the European Cancer Information System (ECIS) [2]. This means there is a 1 in 8 chance of developing breast cancer in Spain each year, with the highest incidence in the 45–65 age group [2].

Breast cancer has physical, emotional, and economic effects for women who suffer from it, who worry not only about the disease but also about the impact on their social and family environment and their employment situation. Thus, breast cancer is one of the three main diagnoses of temporary incapacity that exceeds 365 days of sick leave [3]. In fact, in Spain in 2018, of the 12,245 breast cancer processes that initiated temporary incapacity, 9336 reached 365 days of duration (representing a 76.24% of the total sample). Therefore, only 23.76% of the temporary incapacities initiated by breast cancer patients lasted less than 365 days [4]. Similar data were found in a French cohort with 93% of the patients with at least one period of sick leave and with two periods and 186 days of sick leave on average [5].

Currently, the prognosis for survival of breast cancer in Spain is around 87% in women of working age. However, despite the favorable prognosis of the disease, only 53% have returned to work [4].

Indeed, breast cancer can have a significant economic impact on individuals and health systems. The direct costs of treating breast cancer, including medical expenses such as surgery, chemotherapy, and radiation therapy, can be substantial. Nonetheless, there are also high indirect costs associated with the disease, such as lost income due to missed work or reduced productivity (i.e., due to employment change, absenteeism, and presenteeism of patients and caregivers) [6].

Estimates of indirect costs of breast cancer and absenteeism range from $8068 to $21,086 per patient per year [7]. Particularly, it has been estimated that, per woman, the value of productive days lost (at work and at home) due to metastatic breast cancer ranged from $680 for older women to $5169 for younger women [8]. Even though the absenteeism costs of other diseases such as the COVID-19 might represent a significant economic burden (estimated at nearly $1.3 million, with an average of $671 per patient among the health care workers); as seen, the costs of absenteeism in breast cancer.is much higher compared with those from other common diseases [9].

Return to work may be limited not only by the impact of the disease on women’s quality of life (QoL), but also by the effects of the treatment itself [10]. However, return to work (RTW) is beneficial both for the women themselves and for society. For the social security system, it is imperative for economic reasons to encourage patients to return to work whenever possible [11].

Socially, work is presented as a source of QoL that helps to re-establish identity and social relations for women with cancer [12]. Similarly, work provides a sense of normality and a future, allowing patients to feel again part of a society to which they contribute significantly [13].

Quantitative approaches have focused on the experience of women suffering from breast cancer based mainly on questionnaires, revealing the beneficial effect of returning to work on the health and well-being of these women [14,15,16,17,18,19]. However, there is a need to include qualitative analyses to show the real experience not only of breast cancer survivors, but also of two other social actors, such as health professionals and company managers.

Therefore, this manuscript aims to identify the factors that may act as barriers or facilitators of the RTW of women who have overcome breast cancer in Spain from a qualitative perspective. This study broadens the perspective of the patient to include that of the healthcare professionals involved in the cancer diagnosis and treatment process, as well as company managers.

## 2. Materials and Methods

A qualitative research study using three different “focus groups” (formed by health professionals (n = 8), working women breast cancer survivors (n = 6), and company managers (n = 7), Table 1, was carried out between March and December 2021 in Zaragoza; the fifth city in Spain in terms of population with 700,000 inhabitants [20].

The design of the qualitative study followed the “COREQ” criteria (Consolidated criteria for reporting qualitative research) criteria for qualitative research reporting [21].

The data was managed using Nvivo version 12 qualitative software [22].

### 2.1. Study Participants

Participants were asked to sign an informed consent form that was sent previously regarding their participation in the project. The recruitment process was as follows:

Health professionals related to breast cancer of the main hospitals of Zaragoza were contacted through their Hospital Services/Departments while breast cancer patients were directly recruited by a nurse of the Breast Cancer Unit.

Employers were contacted through the Heads of Human Resources (HR) or Occupational Risk Prevention Departments of companies belonging to the Confederation of Employers and the Confederation of Small and Medium Enterprises in Aragon.

The inclusion criteria for health professionals were: (i) to be a specialist in areas directly related to the treatment and follow-up of women with breast cancer and (ii) to have >5 years’ experience.

The inclusion criteria for working women breast cancer survivors were: (i) women with breast cancer between 18–65 years old who agreed to participate; (ii) active employees at the time of diagnosis; (iii) with different medical treatments. Main characteristics are detailed in Table 2.

The following inclusion criteria should be met by employers: (i) to have women diagnosed with breast cancer among their workers; (ii) willing to participate; and (iii) an interest in supporting medical and scientific research in this field.

The characteristics of the representatives of companies that had experience in caring for workers diagnosed with breast cancer and whose responsibilities are detailed in Table 3.

### 2.2. Data Collection Procedures

The outbreak of coronavirus disease necessitated the use of virtual platforms (Google Meet^®^ in the group of health professionals and Zoom^®^ for patients). The group of company managers could be conducted in person. Discussions were facilitated by a Core moderator with extensive experience in facilitating focus groups. Each focus group included 6 to 8 participants and lasted approximately 90 min and they were conducted by two trained moderators in qualitative studies.

To facilitate discussion, the interviewers used open-ended questions based on the literature and on the previous experiences with women with breast cancer if this information had not been spontaneously mentioned by the woman.

Participants were recorded in an audio file, with the prior consent of the participants, to guarantee adequate data processing, which was subsequently transcribed. Transcripts were analyzed using inductive content analysis to identify work-related barriers and facilitators and coded by the research team. This provided the initial coding scheme, which included the themes and their definitions.

To ensure the reliability the Grounded Theory of Strauss and Corbin was used to analyze step-by-step the data [23]. During the interviews, patterns that seemed to emerge were incorporated into the subject guidelines. The participants’ speech was converted to text and analyzed. Substantial phenomena were named, and codes were written in the margins next to the relevant data. In this way, substance was extracted from the data (substantive codes). Similarities and differences in the data with the same code were compared changing or creating new codes. Similar events were then grouped into larger parent groups and given names and categories (tentative categories). Each preliminary category was developed based on open coding to identify the quality in the data.

Theoretical considerations were made throughout the analysis. All interviews were conducted by the same researcher. To ensure validity, two investigators were continuously involved in the coding process. The researchers tried to ensure the validity of the results by asking the participants. Notes were also written during the process to ensure that the researcher’s impressions, thoughts, and reflections were not lost during the analysis. In addition to recording interviews, the authors observed the participant’s posture, body communication, and facial expressions, as well as their verbal communication patterns. These observations were recorded as part of the process and could then be compared with the processed raw data.

### 2.3. Ethical Considerations

The study was developed in compliance with the ethical principles of the 1964 Declaration of Helsinki, revised in 2000 in Edinburgh. The standards of good clinical practice of the International Conference on Harmonization for Good Clinical Practice were respected [24]. All participants who wished to participate in the study signed an informed consent form.

This project was approved by the Research Ethics Committee of the Autonomous Community of Aragón, C.P.-C.I. P119/326 Acta No 21/2019.

## 3. Results


1. Health professionals:


13 open questions grouped into three dimensions: the impact of the disease on employment, the impact of the clinical features and therapies applied on the patients’ RTW, and the contributions and coping of the health care staff, were discussed in this focus group (Appendix A).

The experiences and opinions were clustered in these following four themes:


*The impact of breast cancer on women’s employability from the perspective of health care professionals.*


Health professionals stated there was no specific training on the importance of returning to work and its positive effect on health and QoL of the patients although they indicated workers who suffered breast cancer had a higher rate of unemployment.

The group stated that, although the RTW was very positive in the recovery process of the woman, the limitations derived from the treatment make it necessary to maintain the situation of incapacity to work:


*“Work is what works best for depression. However, there are patients who find it difficult because of the working conditions, …, they ask us for reports, and we have to try to help them facilitating that incapacity”.*

*(Psychiatry specialist)*



*Factors acting as barriers to RTW*


The group pointed out these factors limiting the RTW: (i) the type of activity performed, with jobs with greater physical requirements and lower qualification those that presented more difficulties for RTW; (ii) advanced stages of cancer; (iii) those requiring more aggressive treatments; and (iv) low educational level of the patient.

The mental health status of the worker also appeared as a determining factor in the process of returning to work:


*“The psychological sequelae derived from the disease make up the main barrier to RTW, being, moreover, an aspect in which it is difficult to intervene”.*

*(Specialist in occupational medicine)*


The fear that the diagnosis itself or the need for a new sick leave would mean a stigmatization of women in the company is pointed out as a barrier:


*“(…) Adaptation is beneficial even for the company itself… That gives a very good image, and that good image is what companies also want to have”.*

*(Occupational medicine specialist)*



*Factors that behave as facilitators for the Return to Work*


The motivation of each person to RTW was found as a key aspect:


*“Even work can be an incentive for recovery, something to which patients cling to in order to overcome the disease…” .*

*(Gynecology specialist)*


The importance of social support for the patient was highlighted by several specialists:


*“I think it is important to always ask the patient how she is doing, what family support she has, because that can completely change the prognosis of how she will evolve”.*

*(Surgical specialist)*



*Factors related to discharge status and collaboration between health professionals.*


Some specialists suggested the need for a gradual RTW after discharge, to facilitate a balance between the patient’s psychophysical condition and the requirements of the job.

The specialists in Occupational Medicine did not agree with the viability of this measure:


*“In practice…that is a chimera. However, this gradual reincorporation does not usually take place. Progressive adaptation does not usually exist in private companies”.*

*(Occupational medicine specialist)*


Also, it was pointed out that the prescription of sick leave should be granted not only by specialists in Family Medicine but also by Surgeons, Oncologists or Gynecologists, since they are the ones who really know the patient’s state of health.

The importance of multidisciplinary collaboration was unanimously underlined and considered essential. However, Occupational Medicine physicians manifested they are excluded from the health system not having access to all data related to treatment and clinical evolution of the patient making that collaboration difficult.


*“When we receive information from other specialties, we often find that it is very general and does not usually include details that are nevertheless necessary for the adaptation of the position in practice”.*


Appendix A summarizes the opinions and perceptions expressed during the discussions in the focus group of health professionals in relation to the impact at the time of diagnosis, as well as the barrier and facilitating factors that were presented during the session.


2. Working women breast cancer survivors


A total of 20 issues were discussed in this focus group (Appendix A) regarding the impact of breast cancer on their working life at three clinical stages of the disease: at diagnosis, during treatment (the treatment and the relationship with the health professionals and the sick leave or temporary incapacity) and at the end of treatment (barriers and facilitating factors to RTW).


*Situation at diagnosis*


The word “cancer” generated an intense feeling of uncertainty in the women:


*“…The world falls on top of you because it is like looking into an abyss. You don’t know what you are going to find; you know that everything is going to change…”.*

*(Worker with breast cancer)*


The way of facing the disease varied from each person, showing a greater or lesser degree of optimism depending on the personality. However, as the process progressed, the treatment was internalized and assumed. Work was not one of their main concerns, but rather it was the response to treatment and the side effects derived from it that were among their greatest worries.

Most of them shared the diagnosis with their work colleagues and superiors, although they expressed the difficulty of verbalizing it; only one of the patients, who worked in the public system, did not report the disease in her work environment. After communicating it in their company, most of them felt supported.


*Situation during treatment and relationship with health professionals*


The participants rated as “exceptionally good” the treatment received by all health professionals.

They valued empathy positively, particularly during the diagnosis. One of the participants, who had not yet completed the treatment and had not been able to return to her job, acknowledged that it was an extremely complex period, requiring antidepressant medication. The side effect of hair loss was particularly complicated for her:


*“It’s like losing your personality”.*

*(Worker with breast cancer)*



*Situation during treatment: sick leave or temporary disability*


The approximate duration of sick leave for the women participating in the study who had overcome the disease and returned to work was between 10 and 18 months. However, due to the COVID-19 pandemic the plans of some of them at the beginning of the confinement were modified.

The sick leave was shorter in one of the participants because she could “telework”. Although she experienced greater fatigue, she preferred to continue working”.

One of the participants who had already returned to her job, shared her experience:


*“I had a terrible time during my sick leave, and it is true that while I was undergoing radiotherapy treatment, I thought it was fine to be on sick leave. But the confinement arrived…and I talked to my colleagues, who were there working from eight to eight in the eight (…) So, I felt very bad, not being there with them”.*

*(Worker with breast cancer)*



*Situation at the end of the treatment: barriers to RTW*


One of the main barriers identified by the participants was the feeling of “depersonalization” of the worker in large companies. One of the participants was dismissed following her sick leave, but after being declared “unfair dismissal”, the company facilitated her reintegration.

In another case, after returning to work, she was transferred to another department to perform an activity that required lower qualifications.


*“It’s not enough that I recovered from cancer, but I came back to the company, and they told me: you’re out and you’re going to start from scratch”.*

*(Worker with breast cancer)*


The main sequelae of the treatment of the disease mentioned by patients who have returned to work, and which may limit their activity were:

(a) Difficulty in moving around and pain on the same side as the mastectomy.

(b) Lack of concentration and memory secondary to chemotherapy or hormone therapy, both of which generate greater insecurity in the performance of their tasks.


*Situation at the end of treatment: facilitating factors for labor reinsertion and integral recovery*


The participants agreed on the need for a protocol to improve physical recovery, based on a reinforcement of the rehabilitation treatment. They considered physical exercise as a tool to improve their situation.

Another aspect underlined by the patients was the need to know more accurately the waiting times for completing disease-related interventions (i.e., breast reconstruction).

They also considered that knowledge of the disease by the companies could favor support for patients and the possibility of having legal advice would facilitate the knowledge and defense of their rights.

The support of colleagues and superiors and overall family was perceived as a factor that favors the RTW and recovery.

Finally, one of the interviewees pointed out that it could be useful to increase the tools for the early diagnosis of cancer in company examinations.

Appendix A summarizes the opinions and perceptions expressed by women workers diagnosed with breast cancer in relation to the impact at the time of diagnosis, as well as the barrier and facilitating factors.


3. Business leaders: development of the focus group


A total of 12 issues were discussed in this focus group which were grouped into: (i) the diagnosis of cancer in the employee and its impact on the company and (ii) tools to improve the process of reintegration into the workplace. Appendix A shows the themes and issues discussed in the focus group session of health professionals.


*The employee’s diagnosis of cancer, her perception and impact on the company*


Employers stated, in most cases, employees reported the diagnosis to the HR departments, although it was not mandatory:


*“…She came to talk to me and told me about it calmly. Above all, she was very worried about the time she was going to miss, what the company would think, her colleagues, (…) things that would be on everyone’s mind at the time…”.*


Participants agreed on HR management should consider the emotional impact that a cancer diagnosis had on the workers.


*“There will be sick people who will benefit from going back to work, and others the opposite way”.*


The group agreed about the need to create a legal framework that adapts to different situations and guarantees job stability after the period of temporary incapacity, while giving companies the support to deal with these situations.


*“The peace of mind that nothing is going to happen, neither to me nor to the company, I think that would be a great help to recovery”.*


Another statement was made was that “sick leave must be “normalized”:


*“What is anormal are workers who never take sick leave, and nothing happens to them, but sickness is part of life…”.*



*Tools to improve the RTW*


There was an agreement about creating a climate of trust and communication between workers and HR departments. They thought companies should adapt their schedule and tasks, evaluating their physical and psychological sequelae but some situations make it impossible (i.e., jobs requiring high physical activity), because some companies might not have alternatives to change their jobs.

Employers pointed out that while their companies had a welcome protocol, there was no return-to-work protocol although it would be useful. Moreover, they pointed out the necessary collaboration between the health professionals who follow the patient and the specialists in Occupational Medicine in the company.

Appendix A shows the opinions of company managers at the diagnosis of breast cancer, as well as the barrier and facilitating factors of RTW.

## 4. Discussion

This study aimed to understand the factors that facilitate or impede the RTW of working women who had overcome breast cancer considering the vision of these patients, health professionals and company managers.

Physical fitness and emotional and socio-labor support were seen for all groups as facilitating factors for RTW. Nevertheless, sometimes opinions and experiences expressed by these three groups differed. For example, both groups, health professionals and company managers considered the psychological situation of the patients as a key element in the reinsertion process. Indeed, and in agreement with previous investigations, the group of health professionals highlighted that the psychological sequelae of the disease were the main barrier to RTW [25,26,27]. However, the patients who RTW, although they recognized the emotional impact at the time of diagnosis, patients emphasized the need from a physical rather than psychological recovery. This could be due to the selection of the sample: four of the six women had already returned to their jobs or because emotional management improves progressively throughout the clinical process. In fact, breast cancer has a major impact on working life, and it is not just because of the course of the disease. Women with breast cancer are at increased risk of leaving the workforce due to mental health issues, fatigue and pain-related symptoms, lymphedema, cardiovascular disease, and inflammatory disease [28].

According to previous studies, RTW rates range from 75–85%. However, this percentage seemed to be much lower in other countries such as Spain, in which RTW was reported by only 56% in a Spanish breast cancer using a questionnaire [29].

Regarding the communication of the diagnosis of breast cancer in the company, it was essential for both patients and company managers, the support and empathy. This communication appeared to be simpler in the environment of small and medium-sized companies.

In relation to RTW, patients expressed the lack of need for specific adaptations. Situations of dismissal or non-inclusive treatment were mentioned, with surviving workers being assigned to lower category functions. These statements are coincident with the scientific evidence of higher rates of unemployment in female breast cancer survivors [30]. Nevertheless, company managers expressed that dismissal of these workers was not an option to be considered.

Health professionals pointed out that RTW could contribute to the integral recovery of the patient. In fact, the evidence has identified that work might be the most important aspect related to the improvement in the QoL of these patients [31]. The RTW and adaptation could benefit both the worker, and the company itself, which sees its social image improved.

Multidisciplinary interventions and communication aimed at the comprehensive treatment and recovery of patients, involving also occupational medicine specialists, employers, and human resources professionals, was seen as a positive tool on the RTW process. However, the mechanisms to make such collaboration seemed to be difficult to implement.

Despite the identification of facilitators and barriers in the RTW in this manuscript, it should be bear in mind that these factors affecting the RTW appear to extend over a period of many years after stopping the treatments. The study of Sevellec et al., showed that 6 years after returning to work, even though half of the employees were still working in the same company as before, being diagnosed with cancer, mental health problems and physical issues were still highly prevalent [32].

To our knowledge, this is the first investigation in Spain that has brought together focus groups of working women breast cancer survivors, health professionals and company managers to identify the facilitating and the barriers in the RTW of working women with breast cancer.

We understand that there might be other groups that could also be used in the triangulation process such as the families of the women who had overcome a breast cancer. In addition, the sample was small (N = 21) and purposively selected, which means that it was not expected to be representative of the general population. It should be also noted that advanced disease and chemotherapy are major factors that influence RTW with longer sick leaves [33], so the facilitators and barriers might have been different if we had selected metastatic participants. However, we consider the three focus groups selected in the present study were a well-selected sample sufficiently representative to allow in-depth exploration of the barriers and facilitators in the RTW of working women survivors of breast cancer.

Furthermore, we have to take into account the self-selection bias of participants willing to participate in a group discussion (i.e., except one, all company managers who accepted to participate were women) as features of the limitations of focus group studies in general. In addition, this research was strongly affected by the declaration of the state of alert because of the COVID-19 pandemic, which made the development of this qualitative phase particularly difficult. Due to this situation, two of the discussion groups took place via virtual platforms (Google Meet^®^ in the group of health professionals and Zoom^®^ in the case of the group of patients).

## 5. Conclusions

In conclusion, this qualitative study has provided a list of barriers and facilitators that patients, health professionals and company managers may experience with breast cancer survivors when RTW. Results from this study may help company managers and health professionals to better assist them while facing work related problems derived from the disease and the aggressive treatments.

## Figures and Tables

**Table 1 cancers-15-00874-t001:** Composition of the three focus groups.

Focus Group	Number	Characteristics
Health Professionals	8	Different specialties (medical doctors) with experience in the care of women with breast cancer. 2 specialists in the Occupational Risk Prevention Service: Occupational Medicine with 30 years and 25 years of experience. 1 specialist in Primary Care with 30 years of experience. 1 specialist in Breast surgery with 20 years of experience. 1 specialist in Gynecology and Obstetrics: Breast Unit with 15 years of experience 2 specialists in Oncology: Breast Unit with 15 years of experience. 1 specialist in Psychiatry: Psychosomatic and oncology patients’ unit with >15 years of experience.
Working women breast cancer survivors	6	Working women diagnosed and treated for non-metastatic breast cancer.
Company managers	7	Human Resources or Occupational Risk Prevention Managers of small, medium, and large companies in which there are workers diagnosed with breast cancer.

**Table 2 cancers-15-00874-t002:** Description of the main socio-occupational characteristics of the female workers in the focus group.

Participant	Age at Diagnose	Marital Status	Sector	Seniority in the Position	Type of Company	Return to Work
1	49	Single	Administration	16 years	Big	Yes
2	49	Single	Commercial	14 years	Micro	Yes
3	49	Married	Sanitary	14 years	Civil servant	Yes
4	45	Single	Teaching	6 years	Civil servant	Yes
5	54	Married	Services	15 months	Big (public service, temporary work)	No
6	47	Single	Commercial	19 years	Big	No

**Table 3 cancers-15-00874-t003:** Description of the functions of the people in the group of company representatives.

Participant	Gender	Function	Company Size	Sector
1	Woman	HR Management	Big	Biotechnology
2	Woman	HR Management	Big	Technology
3	Woman	HR Management	Small and medium-sized	Confederation
4	Woman	Occupational Risk Prevention	Small and medium-sized	Metal
5	Woman	HR Management	Small and medium-sized	Engineering
6	Woman	HR Management	Small and medium-sized	Healthcare
7	Man	Occupational Risk Prevention	Big	Transport

## Data Availability

The data presented in this study are available in this article.

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
