# Peer review of "Facilitating Factors and Barriers in the Return to Work of Working Women Survivors of Breast Cancer: A Qualitative Study"

_cancers, 2023, doi:10.3390/cancers15030874_

Round 1

Reviewer 1 Report

This is an interesting and well written paper.

My only concern is about the selection process to form the three focus groups. Since the three groups were independently selected, what population do they represent? This is one of main concerns in some qualitative studies that do not adequately describe the selection process.

Please describe and discuss more extensively the bias originated by the selection process of the focus groups

Author Response

This is an interesting and well written paper.

My only concern is about the selection process to form the three focus groups. Since the three groups were independently selected, what population do they represent? This is one of main concerns in some qualitative studies that do not adequately describe the selection process.

Please describe and discuss more extensively the bias originated by the selection process of the focus groups

Answer: Thank you very much for your comment. We have used the technique called triangulation to increase the reliability and validity of our findings. By using triangulation, we have compared the results obtained from different sources to have a more complete and accurate view of the phenomenon addressed. In this case we have complemented the vision of women who had overcome a breast cancer with the healthcare professionals involved in the cancer diagnosis and treatment process, as well as managers of companies in which there were or had been women workers diagnosed with breast cancer.

We understand that there might be other groups that could also be used in the triangulation process such as the families of the women who had overcome a breast cancer. Also, the sample was small (N=21) and purposively selected, which means that it was not expected to be representative of the general population. However, we consider the three focus groups selected in the present study were a well-selected sample sufficiently representative to allow in-depth exploration of the barriers and facilitators in the RTW of working women survivors of breast cancer.

We have included the following sentences in the limitation section of the discussion as suggested by the reviewer:

“To our knowledge, this is the first investigation in Spain that has brought together focus groups of working women breast cancer survivors, health professionals and company managers to identify the facilitating and the barriers in the RTW of working women with breast cancer.

We understand that there might be other groups that could also be used in the triangulation process such as the families of the women who had overcome a breast cancer. Also, the sample was small (N=21) and purposively selected, which means that it was not expected to be representative of the general population. However, we consider the three focus groups selected in the present study were a well-selected sample sufficiently representative to allow in-depth exploration of the barriers and facilitators in the RTW of working women survivors of breast cancer in Spain.

Furthermore, we have to take into account the self-selection bias of participants willing to participate in a group discussion (i.e., except one, all company managers who accepted to participate were women) as features of the limitations of focus group studies in general”.

Reviewer 2 Report

Studies that work on absenteeism due to diseases are very interesting topics for readers and health policy makers, especially since these absences can significantly reduce the productivity of the workforce.

Absenteeism also imposes the cost of lost productivity due to illness on society. I’ve read with attention this valuable study and offer some comment to improvement the work.

The following sentence needs more clarity:

“breast cancer is one of the three main diagnoses of temporary incapacity that exceeds 365 days of sick leave”

The introduction could be better and it needs more literature review.

It is better to do a literature review on sick-leave and their cost of productivity loss due to breast cancer.

Also, a comparison can be made between sick-leave due to cancer and other diseases such as Covid-19.

For this purpose, you can use the following article:

·         The lost productivity cost of absenteeism due to COVID-19 in health care workers in Iran: a case study in the hospitals of Mashhad University of Medical Sciences

You can also use the following article for the effect of breast cancer on the quality of life of patients:

·         Cost-Utility of "Doxorubicin and Cyclophosphamide" versus "Gemcitabine and Paclitaxel" for Treatment of Patients with Breast Cancer in Iran

In the method, it is well explained how to select the interviewees, but it needs to be explained about how to measure the validity and reliability of the findings.

The type of questions should also be explained. Are the questions open-ended or closed-ended? Was there an interview guide?

State the type of qualitative study and tell in the method what software was used to analyze the interviews.

The discussion has not been able to cover the findings well. This requires a comprehensive literature review and comparison with other studies. The following studies will be useful:

·         Causes of sick leave, disability pension, and death following a breast cancer diagnosis in women of working age

·         Determinants of return at work of breast cancer patients: results from the OPTISOINS01 French prospective study

·         Return to work of breast cancer survivors: a systematic review of intervention studies

Author Response

Studies that work on absenteeism due to diseases are very interesting topics for readers and health policy makers, especially since these absences can significantly reduce the productivity of the workforce.

Absenteeism also imposes the cost of lost productivity due to illness on society. I’ve read with attention this valuable study and offer some comment to improvement the work.

The following sentence needs more clarity: “breast cancer is one of the three main diagnoses of temporary incapacity that exceeds 365 days of sick leave”.

Answer: Thank you very much. We have clarified the sentence and we have added the following paragraph:

“In fact, in Spain in 2018, of the 12,245 breast cancer processes that initiated temporary incapacity, 9,336 reached 365 days of duration, 76.24%; and therefore, only 23.76% of the temporary incapacities initiated by breast cancer lasted less than 365 days [4]. Similar data were found in a French cohort with 93 % of the patients with at least one period of sick leave, with on average 2 period and 186 days of sick leave [5]”.

The introduction could be better and it needs more literature review.

Answer: Thank you very much for your suggestion. We have tried to improve our paper thanks to the valuable reviewer´s suggestions. All the changes have been highlighted using track changes

It is better to do a literature review on sick-leave and their cost of productivity loss due to breast cancer.

Answer: Thank you very much for your comment. We have included some literature review on sick-leave and their cost of productivity loss due to breast cancer in the introduction section.

“Indeed, breast cancer can have a significant economic impact on individuals and health systems. The direct costs of treating breast cancer, including medical expenses such as surgery, chemotherapy, and radiation therapy, can be substantial. Nonetheless, there are also high indirect costs associated with the disease, such as lost income due to missed work or reduced productivity (i.e. due to employment change, absenteeism, and presenteeism of patients and caregivers) [6].

Estimates of indirect costs of breast cancer and absenteeism range from $8,068 to $21,086 per patient per year [7]. Particularly, it has been estimated that, per woman, the value of productive days lost (at work and at home) due to metastatic breast cancer ranged from $680 for older women to $5,169 for younger women [8]”.

Also, a comparison can be made between sick-leave due to cancer and other diseases such as Covid-19.

For this purpose, you can use the following article:

  • The lost productivity cost of absenteeism due to COVID-19 in health care workers in Iran: a case study in the hospitals of Mashhad University of Medical Sciences

You can also use the following article for the effect of breast cancer on the quality of life of patients:

  • Cost-Utility of "Doxorubicin and Cyclophosphamide" versus "Gemcitabine and Paclitaxel" for Treatment of Patients with Breast Cancer in Iran

Answer: Thank you for the references and suggestion. We have included the following paragraph:

Estimates of indirect costs of breast cancer and absenteeism range from $8,068 to $21,086 per patient per year [7]. Particularly, it has been estimated that, per woman, the value of productive days lost (at work and at home) due to metastatic breast cancer ranged from $680 for older women to $5,169 for younger women [8]. Even though the absenteeism costs of other diseases such as the COVID-19 might represent a significant economic burden (estimated at nearly $1.3 million, with an average of $671 per patient among the health care workers); as seen, the costs of absenteeism in breast cancer.is much higher compared with those from other common diseases [9].

In the method, it is well explained how to select the interviewees, but it needs to be explained about how to measure the validity and reliability of the findings.

Answer: Thank you very much for pointing this out. We have included the following paragraph:

“To ensure the reliability the Grounded Theory of Strauss and Corbin was used to analyze step-by-step the data [1]. During the interviews, patterns that seemed to emerge were incorporated into the subject guidelines. The participants' speech was converted to text and analyzed. Substantial phenomena were named, and codes were written in the margins next to the relevant data. In this way, substance was extracted from the data (substantive codes). Similarities and differences in the data with the same code were compared changing or creating new codes. Similar events were then grouped into larger parent groups and given names and categories (tentative categories). Each preliminary category was developed based on open coding to identify the quality in the data.

Theoretical considerations were made throughout the analysis. All interviews were conducted by the same researcher. To ensure validity, two investigators were continuously involved in the coding process. The researchers tried to ensure the validity of the results by asking the participants. Notes were also written during the process to ensure that the researcher's impressions, thoughts, and reflections were not lost during the analysis. In addition to recording interviews, the authors observed the participant´s posture, body communication, and facial expressions, as well as their verbal communication patterns. These observations were recorded as part of the process and could then be compared with the processed raw data”.

The type of questions should also be explained. Are the questions open-ended or closed-ended? Was there an interview guide?

Answer: Thank you. We have included the following paragraph in the methods sections:

“To facilitate discussion, the interviewers used open-ended questions based on the literature and on the previous experiences with women with breast cancer if this in-formation had not been spontaneously mentioned by the woman.”

State the type of qualitative study and tell in the method what software was used to analyze the interviews.

Answer: Thank you very much for your comment. We have included the following paragraph:

A qualitative research study using three different "focus groups" (formed by health professionals (n=8), working women breast cancer survivors (n=6), and company managers (n=7), Table 1, was carried out between March and December 2021 in Zaragoza; the fifth city in Spain in terms of population with 700000 inhabitants [21]. The data was managed using Nvivo version 12 qualitative software [22].

The discussion has not been able to cover the findings well. This requires a comprehensive literature review and comparison with other studies. The following studies will be useful:

  • Causes of sick leave, disability pension, and death following a breast cancer diagnosis in women of working age

  • Determinants of return at work of breast cancer patients: results from the OPTISOINS01 French prospective study

  • Return to work of breast cancer survivors: a systematic review of intervention studies

Answer: Thank you very much for the references and suggestion. We have completed and improved the discussion section.

Round 2

Reviewer 2 Report

The authors have clearly improved the manuscript and made it really clearer. It is suitable for publication.